# Organic Pollutants Associated with Plastic Debris in Marine Environment: A Systematic Review of Analytical Methods, Occurrence, and Characteristics

**DOI:** 10.3390/ijerph20064892

**Published:** 2023-03-10

**Authors:** Hongrui Zhao, Ileana Federigi, Marco Verani, Annalaura Carducci

**Affiliations:** Laboratory of Hygiene and Environmental Virology, Department of Biology, University of Pisa, Via S. Zeno 35/39, 56127 Pisa, Italy; hongrui.zhao@phd.unipi.it (H.Z.); ileana.federigi@unipi.it (I.F.); marco.verani@unipi.it (M.V.)

**Keywords:** microplastics, organic pollutants, beach, seawater

## Abstract

Plastic pollution has become one of the most serious environmental problems, and microplastics (MPs, particles < 5 mm size) may behave as a vehicle of organic pollutants, causing detrimental effects to the environment. Studies on MP-sorbed organic pollutants lack methodological standardization, resulting in a low comparability and replicability. In this work, we reviewed 40 field studies of MP-sorbed organic contaminants using PRISMA guidelines for acquiring information on sampling and analytical protocols. The papers were also scored for their reliability on the basis of 7 criteria, from 0 (minimum) to 21 (maximum). Our results showed a great heterogeneity of the methods used for the sample collection, MPs extraction, and instruments for chemicals’ identification. Measures for cross-contamination control during MPs analysis were strictly applied only in 13% of the studies, indicating a need for quality control in MPs-related research. The most frequently detected MP-sorbed chemicals were polychlorinated biphenyls (PCBs), polycyclic aromatic hydrocarbons (PAHs), and organochlorine pesticides (OCPs). Most of the studies showed a good reliability (>75% of the total score), with 32 papers scoring 16 or higher. On the basis of the collected information, a standardizable protocol for the detection of MPs and MP-sorbed chemicals has been suggested for improving the reliability of MPs monitoring studies.

## 1. Introduction

Plastic is widely used in various industries and in human daily necessities because of its low price, versatility, flexibility, and durability. As an indispensable product in human daily life, global plastics production reached approximately 370 million tons (Mt) in 2019 [1]. In a 65-year period (from 1950 to 2015), around 6.3 billion tons of plastic wastes were generated, and approximately 80% has been stored in landfills owing to its low-recycling or reusing rates (around half of plastic products are designed for single use, such as utensils, bags, and packaging [2]) or directly released into natural ecosystems as a result of insufficient management practices [3,4]. The discharged plastics and their fragments are transferred to the marine environment via atmospheric emissions, rainwater runoff, drainage systems, and riverine outputs [5,6,7]. A forecasting study on plastic in the environment predicts that the quantity of land-based plastic wastes entering the ocean will increase tenfold by 2025 if appropriate waste management measures are not implemented [8]. Therefore, the marine environment represents the final destination of various plastic polymers, mainly polyethylene (PE) (e.g., plastic bottles, bags, and pipes), polypropylene (PP) (e.g., straws, bottle caps, and milk bottles), polyvinyl chloride (PVC) (e.g., drainage pipe, window frames, and water service pipes), polystyrene (PS) (e.g., food containers, packaging foam, toys), and polyethylene terephthalate (PET) (e.g., microwavable packaging, carbonated drink bottles, furniture, and pillows) [8,9,10].

Generally, tiny plastic fragments are categorized as microplastics (MPs, size between 100 nm and 5 mm) that can be divided into two categories: primary MPs and secondary MPs. Primary MPs are produced in small sizes for commercial use (e.g., cosmetics, textiles, and medicines). Secondary MPs derive from larger items as a result of environmental degradation processes [11,12].

The ingestion of MPs by marine organisms has been reported at various levels of the trophic web, starting from zooplankton taxa that graze MPs alongside other food sources [13] to the higher trophic consumers feeding on contaminated zooplankton, such as bivalves [14] and planktivorous fish [15]. The marine trophic transfer of MPs has been empirically demonstrated from the quantification of MPs in top predators, such as tuna and swordfish [16], as well as baleen whales, that filter a wide size range from plankton up to small fish [17]. Therefore, the contamination of seafood can pose a potential threat also to human health through dietary exposure [18].

In plastic materials, polymerized monomers are combined with chemical additives (i.e., antioxidants, flame retardants, stabilizers, and plasticizers) whose aim is to improve the physical and chemical properties of the products [19]. Once in the environment, plastic products can release their additives as a result of weathering effects, thus representing one of the most significant sources of these compounds in the marine environment [20]. 

Moreover, MPs have a large surface area to volume ratio, and their altered crystallinity, hydrophobicity, and functional groups confer a high adsorption potential for chemical substances from surrounding waters [21]. Some lab-scale and field studies showed such MPs behavior toward a wide range of chemical contaminants, both hydrophobic organic chemicals (e.g., polychlorinated biphenyls—PCBs, polycyclic aromatic hydrocarbons—PAHs) and hydrophilic organic contaminants (e.g., perfluoroalkyl and polyfluoroalkyl substances—PFASs) [22,23]. 

The adverse effects of these substances on human and animal health are well known [24]; therefore, the characterization of the role of MPs as a vehicle of such organic pollutants is important in a One Health perspective, which requires an integrated view of pollutants circulation in different matrices and contexts [25]. 

This paper aims to understand the current degree of knowledge on MPs and their adsorbed organic pollutants in the marine environment using a systematic review approach. Since the comparability of findings from different studies strongly depends on the technical protocols and analytical methods, the present paper has also the specific objectives of (i) comparing the methodologies used for the detection of MPs and MP-associated chemicals across the reviewed studies; (ii) evaluating qualitatively the reviewed studies through a scoring approach; and (iii) proposing a standardizable protocol for sample collection and the analysis of MPs and MP-sorbed chemicals in marine environments. 

## 2. Materials and Methods

### 2.1. Literature Search Strategy

The systematic review was conducted in accordance with the Preferred Reporting Items for Systematic reviews and Meta-Analyses (PRISMA) guidelines [26]. The literature search was conducted on 30 December 2022 using three online databases (PubMed, Web of Science, and Scopus) without time limitations. The query used for the search included (microplastic OR plastic debris) in combination with the following keywords: HOCs, POPs, PAHs, PAEs, EDCs, pesticides, personal care products, per- and polyfluoroalkyl substances, hydrophobic chemicals, additives, plasticizer, vehicle, and marine. The database search provided 11,775 hits: 6013 hits from Pubmed, 1337 hits from Web of Science, and 4425 hits from Scopus. These articles were collected and duplicates were removed. The papers were screened based on the title and abstract for three criteria and were required to meet each of them for the inclusion: (i) monitoring studies focusing on organic pollutants adsorbed by MPs in either coastal areas or on sandy beaches; (ii) peer-reviewed research articles or technical reports providing monitoring data; and (iii) written in English. The paper selection was performed by two researchers. Papers that were considered relevant were retained for a full reading. A total of 60 original articles were read in full and then 20 papers were removed with reason, as reported in Appendix A. In total, 40 studies met the selection criteria and were included in this review [23,27,28,29,30,31,32,33,34,35,36,37,38,39,40,41,42,43,44,45,46,47,48,49,50,51,52,53,54,55,56,57,58,59,60,61,62,63,64,65] (Figure 1). 

### 2.2. Quality Assessment of the Reviewed Papers

The Criteria for Reporting and Evaluating ecotoxicity Data (CRED) methodology, with some adaptations [66,67], was used for evaluating the reviewed papers, as well as in terms of their overall usability (strengths and weaknesses), for establishing a standardizable protocol.

In particular, the quality assessment method was divided into seven evaluation criteria (see Table 1 for their description). Each criterion was scored from 0 to 3, depending on the degree to which the specific criterion was met: 0 = the reported information is very lacking, with a very low usability for developing a standardizable protocol; 1 = only a small part is usable, but the vast majority needs to be improved or additional information is needed; 2 = mostly reliable, but contains unusable parts; and 3 = high reliability and has a high usability. The global score (minimum = 0, maximum = 21) reflects the reliability of these studies and is not a judgment of the scientific merit or contribution of the papers. Papers that obtained 75% or more of the total score were considered good for reliability [68].

## 3. Results and Discussion

### 3.1. Publication Trends in the Field

The included papers (*n* = 40) are summarized in Appendix A. Their time-trend shows a strong increase after 2017 (Figure 2) and notably in 2019 when the World Health Organization published a report entitled MPs in Drinking Water [70], which plausibly attracted greater attention from researchers worldwide. Therefore, the overall trend of interest in this topic is on the rise and the decline during 2022 (only two papers) may be attributable to the low productivity of researchers during the COVID-19 pandemic, especially in fields which rely on physical laboratories and time-demanding experiments (e.g., biochemistry, biosciences, chemistry) [71,72] as well as field sampling-based data collection [73]. Regarding the location of the reviewed studies, most of them were carried out in China (12 papers, 30%), followed by Japan (7 papers, 17.5%), and Spain (5 papers, 12.5%) (Figure 3).

### 3.2. Methodologies for the Analysis of MP-Associated Chemicals

The determination of environmental MPs (and associated chemicals) relies on three main stages: collection, extraction, and identification. The reviewed studies are compared in each methodological stage, as detailed in the following Sections.

#### 3.2.1. Sampling Sites and Frequency of Sampling 

The majority of papers chose the beaches (*n* = 31) as the sampling area, followed by river estuary surface waters or seawaters (*n* = 8) and coastal wetland sediment (*n* = 1) (Appendix A). Studies carried out in beach locations frequently used sand and sediment as synonyms for referring to the environmental matrix where MPs were collected. Here, we decide to use the term “sand” for the studies performed along beaches and we limited the term “sediment” to the single study carried out in the wetland area [57], according to the glossary used by the WHO in the recent guidelines for recreational water safety [74]. 

Beach areas were preferred by the vast majority of the papers (31/40), probably due to the easy accessibility of such areas [75]. Studies that selected beaches as sampling locations typically focused on the intertidal zone (*n* = 28), except for one paper that chose the backshore sand surface [39] and two papers [40,41] that did not provide specific information on the sampling area. Estuaries were chosen by the rest of the reviewed papers because riverine outflows play a key role in the dispersal of plastic waste from inland regions to the oceans, with an estimated annual volume of 265,000 tons [51]. 

Regarding the frequency of sampling, the reviewed studies sampled only once (a single monitoring campaign), except for three studies that performed multiple monitoring campaigns (two [42,63] and five [56]), and one paper that collected samples from 1984 to 2008 (roughly every 5 years) and demonstrated that the concentrations of all detected persistent organic pollutants (POPs) decreased over the last two decades [31]. The results obtained from a single monitoring campaign could produce relatively large errors in the estimation of the actual pollution level in the MPs because the findings can be influenced by several variables, such as the sample size and sampling time. As an example, PE plastic requires more than one year to equilibrate with PCBs in the surrounding environment [76], thus a single sampling at the beginning of this period could underestimate this type of pollution. 

#### 3.2.2. Collection Method, Sample Amount, and Storage

The collection methods showed considerable differences among the reviewed studies, from sampling techniques to sample storage tools (Appendix A), depending on the type of the environmental matrix (sand/sediment or water). The sampling strategies can be classified into three categories: (i) selective sampling, (ii) bulk sampling, and (iii) volume-reduced sampling [77]. Some studies used forceps/tweezers/spoons or by hand to directly collect distinguishable MPs (*n* = 17, selective sampling), while others collected the whole volume of the sample (not only MPs) in containers (i.e., stainless-steel shovel or bags) and used further laboratory approaches to isolate and extract MPs (*n* = 6, bulk sampling). Other studies used sieves/manta trawl/plankton nets on site to reduce the volume of the bulk sample; then, the samples were placed in containers and carried to the laboratory for storage (*n* = 16, volume-reduced sampling). Only one study did not mention the method of sampling and the instrument used to collect the samples (*n* = 1, [41]). 

The studies on sand samples showed a wide heterogeneity in the size of the sampling quadrats for delimiting the sampling area (i.e., 0.25 m × 0.25 m, 0.4 × 0.4 m, 0.5 × 0.5 m, 1 m × 1 m), as well as in the sampling depths (i.e., 1 cm, 2 cm, 0–2 cm, 2–3 cm, 10 cm, and 20 cm) (Appendix A). The MPs concentrations in sand can be influenced by sampling depth [78,79] because the distribution of MPs concentrations is not homogeneous [80]. Besley et al. (2017) [79] discovered large variations in MPs abundances collected from different depths of the beach surface sand, with a greater MPs abundance in the upper 1–5 cm compared to the upper 2 or 10 cm. 

The studies on water samples differ in terms of the size of trawl mesh (i.e., 300 μm, 330 μm, 333 μm, and 500 μm) and of the mouth size of the manta trawl or plankton nets (i.e., 100 × 50 cm, 90 × 15 cm, 90 × 45 cm, 55 × 25 cm, 30 × 15 cm) (Appendix A). There was no standard for the size and material of the trawl used, and only two studies mentioned that their nets were made of silk [43,56]. The sampling methods can influence the abundance and size of the collected MPs, thus determining variations in the concentration of the detected chemicals [81,82].

Overall, the amount of the environmental matrix (sand/sediment or water) was not carefully considered in the reviewed studies, thus they differ greatly in terms of the weight or volume of the collected sand or water, respectively (Appendix A). The environmental matrix was sampled mainly with the aim of reaching a sufficient MPs weight for chemical analysis, and also, in this case, the studies showed great heterogeneity in the quantity of the MPs analyzed (Appendix A), with some of them using a very small MPs amount (e.g., 2–10 MPs pellets, [27]). For the same reason, environmental matrix features (e.g., sand density, sand moisture content, water salinity) have not been recovered by the authors, and the lack of this information does not allow for inferring the possible role of the environment in affecting MPs abundance.

The storage of samples was considered in order to avoid cross-contamination and to allow the preservation of MPs and their adsorbed chemicals (Appendix A). During sample transport from the collection site to the laboratory (short-term storage), general precautions were applied in the reviewed papers, such as transferring the samples to non-plastic (e.g., glass) containers or wrapping them in aluminum foil for light avoidance. If long-term storage was required, the storage conditions included keeping at room temperature, refrigeration at 2–4 °C, and freezing (with a wide range of low temperatures, from −2 °C to −30 °C), but the most frequently used condition was freezing at −20 °C (*n* = 14).

#### 3.2.3. Sample Pretreatment (Digestion) 

One of the major hurdles in MPs studies is the recovery efficiency from organic-rich environmental samples [77]. In general, if MPs are extracted from biofilms, biological tissues or other organic-rich samples (e.g., algae, zooplankton, phytoplankton), a digestion step (e.g., applying chemicals or enzymes) is recommended to dissolve the organic matter in the sample when small MPs (<300 μm) are assayed [67,77,83] and hydrogen peroxide (H_2_O_2_)-based digestion is commonly used [78,84]. Only four of the reviewed studies [50,56,57,58] performed sample purification based on H_2_O_2_ as the reagent (Appendix A). Some authors demonstrated the effectiveness of H_2_O_2_ in purifying MPs without significantly interfering with the shape and dimension of MPs as well as the infrared spectra [85,86], but the effect of H_2_O_2_ pretreatment on the concentrations of MP-related chemicals has been addressed only by a few studies. Mai et al. (2018) [43] examined the effect of H_2_O_2_ digestion on the concentration of MP-associated PAHs, and their results suggested that the H_2_O_2_ method was able to digest the organic matter without significantly affecting the PAHs quantification. However, most papers (*n* = 36), including [43], did not ultimately apply the pretreatment method. In fact, the omission of the digestion step could be reasonable in case of MPs larger than 300 µm because the residual organic matter does not hamper their identification through Fourier transform infrared (FTIR) spectroscopy [43,86]. Since there is a lack of information regarding the possible impact of digestion treatments on the concentration of organic pollutants related to MPs, the digestion step has not been included as a criterion for evaluating the reviewed papers.

#### 3.2.4. Density Separation and Extraction

The separation of MPs from the environmental matrix depends on the type of sampling collection method. Approximately 73% of the reviewed studies (*n* = 29, Appendix A) did not report the methods used for the separation of MPs from the collected environmental sample, either because they had already obtained a sufficient quantity of MPs through the sampling tools (e.g., trawl screening or direct pick-up with forceps) or because they manually collected plastic debris that were easily distinguishable (e.g., resin pellets). 

In case of water matrices, the MPs extraction is simultaneous to the sampling since filtration (e.g., manta ray trawl with different mesh counts) is able to separate MPs from the water. In the case of sediment sampled by selective (*n* = 17) or volume-reduced (*n* = 16) procedures, the MPs were physically separated during sampling, namely, they were recovered by hands (selective sampling) or through the use of sieves (volume-reduced sampling). The sieve allows MPs separation on the basis of their dimension, and studies widely differed in the mesh sizes (i.e., 0.6 μm, 20 μm, 125 μm, 250 μm, 0.3 mm, 0.4 mm, 0.7 mm, 1 mm, 2 mm, 3 mm, 5 mm, 20 mm; *n* = 12) as well as in the pore size of the filter membranes (i.e., 20 µm nylon filter, 0.45 µm nitrocellulose filter, 0.45 µm, 0.6 µm and 0.7 µm glass microfiber filter; *n* = 6) (Appendix A). Therefore, the extraction method can influence the quantity and size of the recovered plastic particles. For example, Fries et al. (2013) [33] used a filter with a pore size of 0.45 µm, which was an order of magnitude smaller than the filter used by Deng et al. (2021) [57] (20 µm). Therefore, a higher abundance of MPs may be detected using smaller pore sizes, while larger pore sizes can determine the loss of tiny MPs.

During bulk and volume-reduced samplings, the collected samples contain various impurities that could interfere with the subsequent phases of extraction and identification of MPs [78]. Therefore, eleven studies separated MPs from impurities (e.g., sand, shells) on a density criterion through the use of different liquid solutions: sodium chloride—NaCl (*n* = 7), lithium metatungstate—Li_2_O_13_W4^−24^ (*n* = 1), sodium iodide—NaI (*n* = 1), ambient seawater with an addition of NaCl (*n* = 1), and distilled water (*n* =1). The rationale behind the density separation is that MPs float on the surface of the solution [83], but the type of recovered MPs depends on the density of the solution. Water (distilled or freshwater) was used for the separation of lightweight foams with densities less than 1 g/cm^3^ [53,83], while a saturated NaCl solution (density 1.2 g/cm^3^) is suitable for higher density polymers to avoid an underestimation of the total quantity of plastic [83,87]. However, some plastic polymers are close to 1.7 g/cm^3^ (e.g., polyvinyl chloride—PVC, 1.16–1.58 g/cm^3^; polyformaldehyde—POM, 1.41–1.61 g/cm^3^, and polyethylene terephthalate—PET, 1.38–1.43 g/cm^3^); thus, NaCl solution could also not provide a good separation. In those cases, a high-density floating solution was applied, such as Li_2_O_13_W4^−24^ (density = 1.62 g/cm^3^) [44] and NaI (density = 1.8 g/cm^3^) [58,69,88], although their use was limited due to toxicity and costs [69]. NaCl solution was the most commonly used solution for separating MPs since it is environmentally friendly and cheap compared to other chemical solutions [77,83,87]. Additionally, the use of distilled water guarantees laboratory safety and a low cost, but its ability to recover high-density plastics is extremely low; only Shi et al. (2020) [53] used distilled water and they recovered MPs fragments (60%), foam (27%), and pellets (13%); therefore, a possible underestimation of high-density MPs cannot be excluded. 

Therefore, the selection of the separation solution needs to be a good compromise between the polymer types of the searched MPs and the degree of hazard to the environment and human health. Moreover, MPs in the environment can change their density due to the adsorption of contaminants, thus causing impairments in MPs quantification through density separation techniques [83]. This further aspect needs to be carefully addressed in the choice of the separation solution. 

#### 3.2.5. Identification of Polymers 

The MPs extracted from environmental samples can be identified through various methods. The simplest method is the visual inspection, using the naked eye or a microscope, that allows for subjectively determining the appearance (e.g., color, shape, structure, elasticity, or hardness) of suspected MPs (the term “suspected” need to be used when MPs are visually identified, without the use of dedicated instruments). In particular, it is suitable for recognizing plastic resin pellets with a size ranging between 1 mm and 5 mm and unique shape (generally cylindrical or disc-shaped), or plastic foam with a softer texture and low density [27]. Although the visual identification can save time [89], it cannot be used to effectively identify plastic fragments because their shape and color can change due to fragmentation and erosion; moreover, this method can produce error rates of up to 70% because of the subjective nature of visual discrimination [90]. As an example, the color determination of suspected MPs can be prone to error depending on the ability of the researcher to discriminate colors or the background laboratory light or the light used for microscopic analysis [90]. Such limitations can be overcome through the polymer identification of MPs using analytical instruments [90,91] that can verify the real origin of a suspected MPs, also providing information on the possible sources of MPs [83,92].

Of the reviewed papers, 30% (*n* = 13, Appendix A) did not use instrumentations to identify the type of polymer. Therefore, 10/13 studies did not specify the type of polymer, while 3/13 studies reported the type of polymer based on only a visual identification (*n* = 2, naked eye or microscopy [32,40]) or density separation (*n* = 1 [39]). In the study by Van et al. (2012) [32], plastics were classified into pellets, fragments, polystyrene (PS) foam, and rubber based on their physical appearance (color, size, and type). PS foam is an easily recognizable polymer type, but the morphology and color of aging foams can change with the surrounding environment; thus, the polymer information may be inaccurate without an instrumental identification [32]. Similarly, Jang et al. (2017) [40] collected stranded buoys with a clearly identifiable appearance and then they identified extruded PS foam, low-density expanded polystyrene (EPS), and high-density EPS, but without using a methodology to verify their findings. Fisner et al. (2017) [39] recovered PE and polypropylene (PP) pellets using the density separation method (water and ethanol solution) because it showed a good reliability on the basis of the data obtained by Manzano et al. (2009) [93], who compared this method with instrumentation (Raman spectroscopy). However, the density separation method is flawed because some resin pellets are made of PS plastic [28]. 

Although the aim of the reviewed studies was the investigation of MP-related organic chemicals, plastic polymers characterization is crucial for the comparability of the data since different types of polymers can exhibit different absorption properties [94,95]. For this reason, most of the reviewed papers used analytical instruments for an identification of the polymer type (*n* = 27, Appendix A). The large majority of them (25/27) applied spectroscopy techniques; these are nondestructive methods that allow for investigating the specific features of a sample (i.e., consistency or structure) using the infrared light of the electromagnetic spectrum [96,97], such as Fourier transform infrared (FT-IR) spectroscopy, which uses mid-infrared wavelengths of light (*n* = 17), while near-infrared (NIR) spectroscopy (*n* = 7) and far-infrared (FIR) spectroscopy (*n* = 1) use near-infrared and far-infrared wavelengths of light, respectively. Only two papers used spectrometry instruments for MPs polymer analysis, namely, thermal decomposition based on pyrolysis gas chromatography/mass spectrometry (Py-GC/MS), and liquid chromatography coupled to size exclusion chromatography coupled to high-resolution mass spectrometry (LC-SEC-HRMS). The application of different techniques in MPs polymer search is further discussed in Table 2.

#### 3.2.6. Identification of MP-Associated Chemicals 

In the reviewed papers, MP-associated chemicals have been identified through chromatography techniques combined with mass spectrometry for chemicals quantification, using various configurations as detailed in Appendix A. Chromatography is classified into two types based on the physical state of the mobile phase used, gas chromatography (GC) and liquid chromatography (LC). In GC, the sample is vaporized; therefore, it is not suitable for the analysis of thermally labile compounds, whose analysis can be carried out using LC that it is typically performed at room temperature because the sample is dissolved in a solvent. Therefore, the type of chromatography has been chosen on the basis of the features of the searched chemicals. 

Most of the reviewed papers used GC-based techniques (*n* = 32) because the chemicals associated with polymers are mostly stable to a high temperature, are mainly chemicals adsorbed form the surrounding environment (PAHs, PCBs, organochlorine, or organophosphorus pesticides), but are also chemicals which are incorporated into plastic during manufacturing, such as UV filters, phthalates, and PBDEs (polybrominated diphenyl ethers) among flame retardants compounds [106]. The chemicals detection was carried out with mass spectrometry, except for five studies [27,30,37,38,61] that used an electron capture detector as a device for the quantification of some organochlorine compounds, namely, PCBs and organochlorine pesticide (dichlorodiphenyltrichloroethane—DDTs, and hexachlorocyclohexane—HCHs), because it offers a higher sensitivity compared to mass spectrometry [106,107].

The rest of the papers (*n* = 8) used LC-based techniques for the analysis of hydrophilic substances (PFASs) and various hydrophobic plastic additives (UV stabilizers, antioxidants, bisphenol A—BPA, nonylphenols, hexabromocyclododecane—HBCD). In all the reviewed studies, the LC technique was married with mass spectrometry for the quantification.

#### 3.2.7. Contamination Control 

Samples can be contaminated by background MPs and chemicals (e.g., tools, clothing, indoor air) during collection, transport, sample preparation, and analysis. Such external contamination can affect the accuracy of the results; therefore, it needs to be revealed using negative controls or prevented through the application of dedicated control measures [77]. 

Most of the reviewed papers (*n* = 35, Appendix A) performed negative controls to check the possible cross-contamination from: (i) residues of MPs (e.g., airborne MP fibers/debris, contaminated tools, or containers), using a control glass container (e.g., Petri dish) with distilled water (frequently named “procedure blanks” or “field blanks” in the reviewed papers), or (ii) chemicals derived from the instrumental analysis, performing a set of control samples (frequently named “procedure blanks”, “blank controls”, “solvent blanks”). In both cases, negative controls were analyzed simultaneously with the collected environmental samples. As an example, Deng et al. (2021) [57] tested 50 blank samples to assess the background contamination of MPs, showing a presence of very small amounts of MPs in blanks for water (0.28 ± 0.54 items/filter membrane) and sediment (0.56 ± 0.71 items/filter membrane). Although this study was sound in avoiding laboratory contamination during sample processing, a possible contamination could happen during sampling because the authors used plastic buckets for surface water sampling and nylon filters for filtration. However, the study investigated large MPs size (0.33 mm–5 mm); thus, the material of the containers and filter membranes may have less impact on the recovered MPs.

Likewise, control measures against contamination have been frequently applied in the reviewed papers (*n* = 38, Appendix A). In particular, such measures consisted of one (or a combination) of the following approaches: (i) use of non-plastic materials (e.g., aluminum foil) to cover samples, (ii) use of pre-cleaned sampling tools and laboratory equipment (e.g., washing with ethanol, methylene chloride, or deionized water), (iii) work under clean conditions to minimize contamination in the laboratory (e.g., cleaning the laboratory in advance), (iv) wearing clothing and gloves that are non-synthetic or free from the target chemicals, and avoiding contact with any plastic materials (e.g., cotton lab coat), and (v) storing samples in clean containers (e.g., glass vials), or in amber or brown bottles to prevent the possible degradation of the sample. 

However, only 5/38 papers (13%) reported detailed protocols for contamination prevention, as summarized below. In the study of Fries et al. (2013) [33], such measures included the use of aluminum foil, lab coats, and pre-cleaning of the workplace. The authors compared the fibers isolated from the laboratory and from the collected sediments and they found that the background contamination was difficult to avoid; thus, they decided to no longer search the fibers in the environmental samples. Another issue was related to the use of PET bottles rather than non-plastic containers to store the sediment samples, although the authors reported the low risk of background contamination from phthalates in PET drinking water bottles. Zhang et al. (2015) [37] used equipment completely cleaned with dichloromethane (before and after a batch of analyses), and the experiments were performed in a clean laboratory. In addition, they analyzed six blank samples for the target chemicals that were all below the detection limit, thus showing that the applied control measures were effective. Hajiouni et al. (2021) [58] used glass containers, the cleaning of equipment with ultrapure water, cotton lab coats, and nitrile gloves. They also performed negative controls both for possible airborne MPs, placing a glass Petri dish with distilled water on a laboratory cabinet, and for target chemicals, preparing solvent blanks and procedure blanks. They found traces of the target compounds in the procedure blank, so they subtracted such values from the results of the environmental analyses. Capriotti et al. (2021) [55] followed the rules of Baini et al. (2018) [108] for the prevention of contamination, meaning they cleaned sampling equipment prior to use, they used glassware in the experiments, and sample analysis procedures were performed in a clean laboratory fume hood. Moreover, they performed negative controls for external MPs by placing two glass Petri dishes on either side of the microscope. The study by Cheng et al. (2021) [56] had a very detailed procedure for contamination control. To avoid cross-contamination from external MPs, they used nitrile gloves and cotton lab coats, and they cleaned containers and tools with methanol three times until dry and then they wrapped them in aluminum foil. They also paid attention to the cross-contamination from chemicals through the use of materials free from the target compounds during sample collection, transport, and pretreatment, and they removed possible residual compounds in the glass vials using an alkaline solution. Moreover, they performed negative controls for MPs and for target chemicals: most of the blank samples were free from external contamination and those which resulted positive for the presence of textile fibers or target compounds were excluded from the statistical analysis. 

The papers that did not describe any protocols for the contamination control (*n* = 2, [39,62]) were focused on resin pellets and fragments of 1 mm–5 mm; thus, the impact of cross-contamination could be relatively small compared to the fibers search.

Overall, the use of negative controls and the application of control measures against environmental contamination are key procedures to reduce experimental errors. The wide adoption of such procedures in the reviewed papers shows an increase in scientists’ vigilance on contamination risks [78].

### 3.3. Occurrence and Features of Environmental MPs

#### 3.3.1. Differences in Quantification Units

Of the 40 reviewed papers, only 14 studies provided information on MPs abundance (Appendix A). Information on MPs (e.g., abundance, shape, and size) are useful to understand their role as carriers for chemicals and their possible impacts on human/animal/ecosystem health. Nevertheless, the reviewed studies lack uniform detection methods and quantification units for MPs [109] since their targets were MP-associated chemicals. Differences in sampling and identification methods for MPs create differences in the detection limits, with the lower limit of detection depending on the minimum aperture of the filters or membranes [110]. For sediment samples, the quantitative units varied from particles/10 g, items/kg, items/m^2^, to items/m^3^. Instead, MPs from water samples were expressed as ng/mL, particles/100 m^3^, items/m^3^, or items/L (Appendix A). The difficulties in the conversion among different units (e.g., volume, weight, and area) hamper the comparison of MPs abundance data among studies. Therefore, there is a need for standardized protocols for MPs description and investigation (e.g., shape, abundance, aging degree) to improve the reproducibility of the studies that investigate MP-sorbed contaminants. 

#### 3.3.2. Shape, Size, and Color

Once in the marine environment, both primary and secondary MPs are altered in shape and appearance, and the degree of alteration depends on the original shape, the polymer type, the aging processes, and the time spent in the environment [77,103]. The reviewed studies reported a variety of different shapes of MPs, including pellet/sphere/round shape, fragment/sheet/flake, fiber/filament/line, foam, and film (Appendix A). Although foam and resin pellets are not shapes, they have a very recognizable appearance, thus many studies have also included them in the shape classification [78].

Almost half of the reviewed papers (*n* = 16) sampled only resin pellets and did not collect other shapes of MPs (Appendix A). Resin pellets have a high affinity for hydrophobic pollutants, and they are used as POPs monitoring media by various studies [27,30,34]; most of them belong to the monitoring programs named *International Pellet Watch* (IPW; http://pelletwatch.org/). Such plastic particles are industrial raw materials used for the manufacturing of various plastic products [111] and they can be unintentionally released to the environment, both during manufacturing and transport, and may end up in the ocean as a result of surface runoff or river discharges [27]. Endo et al. (2005) [112] indicated that yellowing resin pellets tend to contain higher concentrations of PCBs, probably because they are more persistent in seawater than white/transparent pellets, thus they have more time to absorb PCBs. Another study [27] showed a correlation of PCBs concentrations between pellets and mussels from the same region, thus supporting the use of resin pellets as a monitoring matrix.

The particle size of MPs can influence their ability to migrate in the aqueous environment [113] or to enter the food chain [78,103]. Almost all the reviewed studies reported the size of MPs (*n* = 36) that ranged between 0.25 mm and 5 mm, or they indicated that the size was less than 5 or 20 mm (Appendix A). The selection of relatively large-size MPs can be explained by technical reasons related to chemical analysis, namely, the need of collecting sufficient amounts of chemical contaminants and the better performance of the detection instruments on MPs greater than 0.25 mm. 

As for the particle size, also the color of the MPs can influence the ingestion by animals [78,103]. Approximately half of the reviewed studies (*n* = 21) reported the colors of MPs (Appendix A), and the most frequent were white, black, transparent, red, and blue (Figure 4). The MPs colors can be used to infer their origin from various plastic products [114], but the color of MPs is not permanent, and changes may occur in the environment and during sample preparation [115]. Therefore, caution should be maintained in using color alone to infer the type or source of MPs. For similar colors, it is sometimes very difficult to clearly distinguish them by the naked eye. For example, Hajiouni et al. (2021) [58] grouped indistinguishable colors, such as green and yellow, in the same color category, and white and transparent were classified as one color. 

Both the color and shape of plastic debris can provide a basis for exploring their origin and usage, as well as for evaluating the degree of weathering. For example, virgin foam (used for product shockproofing, insulating packaging, or buoys) is generally white, but aging processes in the environment determines a modification of the color to yellow, gray, or black [78]. In the case of resin pellets, the yellowing is an index for the oxidation of phenolic antioxidants to by-products with a quinone structure [29]. Therefore, pellets with a high yellowness probably stayed longer in seawater, thus providing them with more opportunities to absorb organic pollutants [27,28,112]. For this reason, the reviewed studies on resin pellets tend to select those with high yellowness for the detection of adsorbed contaminants [59]. In the field sampling of resin pellets, the use of a handheld colorimeter would allow for a rapid screening of the aging pellets, thus simplifying the workload of sampling [27].

#### 3.3.3. Polymer Types

The polymer types were reported in 30 of the reviewed studies and the most frequently detected were PE, PP, and PS (Figure 5, Appendix A), which is in agreement with a recent review on MPs in freshwater sediment [78]. Resin pellets were mainly made of PE and PP (78% of the reviewed papers), while other plastic fragments showed various types of polymers (Appendix A). 

Resin pellets are commonly considered the “gold standard” for an MP-sorbed chemical assessment (e.g., POPs) for two main reasons: (i) the aging of resin pellets can be directly verified by colorimetry, given their regular morphologies and color [35]; (ii) the collection of resin pellets is easy because their cylindrical or oblate shape facilitate an identification by the naked eye, and samples can be collected directly without relying on microscopes and additional means (no density separation is required) [28]. Nevertheless, some authors found that other types of plastic debris can also be promising for the detection of MP-associated chemicals. For example, Camacho et al. (2019) [45] found higher concentrations of organophosphate flame retardants in mesoplastics than in pellets. Shi et al. (2020) [53] confirmed the importance of resin pellets to understand the origin and transport of organochlorine pesticides (OCPs), but they also detected higher PAHs concentrations on foams compared to the resin pellets. Further research could be oriented in the search of other type of MP candidates for chemicals detections.

### 3.4. MP-Associated Chemicals

The concentration of the pollutants on plastics in the environment can be influenced by several factors, both related to the MPs (e.g., type, size, and density) and to the chemical pollutants (e.g., hydrophobicity) [27,116]. Moreover, the surrounding environmental conditions (e.g., organic matter, temperature, salinity, pH) can also have an impact on the adsorption of pollutants [116].

The MP-sorbed pollutants most frequently detected were PCBs, PAHs, and OCPs, and usually they were monitored in parallel (Appendix A and Table 3). The number and type of pollutant congeners varied among the reviewed studies, thus hampering the comparisons. Overall, the reviewed studies found a link between the pollutants detected in the MPs and one or more anthropogenic inputs sources, such as urban activities, transport, tourism, agriculture, and industry. 

#### 3.4.1. Polychlorinated Biphenyls (PCBs)

PCBs were widely used for industrial purposes until the 1980s, then they were banned in many countries with the signing of the Stockholm convention [117] because they were recognized as endocrine disruptors, reproductively toxic, teratogenic, and carcinogenic to humans and animals [47,118]. Nevertheless, PCB residues still pose a threat to ecosystems due to their high persistence and their potential of accumulation in living organisms [119,120]. MP-sorbed PCBs were searched in 22 of the reviewed papers (Table 3) with a median concentration of 290 ng/g (Appendix A), with the highest level (2230 ng/g) detected in beach pellet from France [61]. Ogata et al. (2009) [27] found relatively low PCB concentrations (<50 ng/g-pellet) in areas such as Australia, South East Asia, and Southern African countries, where PCBs were banned or not used during the economic boom of the 1980s. In contrast, PCB concentrations were higher along some coastal areas of the United States (300–600 ng/g-particulate) and in Japan and Western Europe (50–400 ng/g-pellet), where PCBs were used extensively in the 1960s and 1970s, thus they strongly accumulated in coastal sediments. Heskett et al. (2012) demonstrated that median concentrations of ∑13 PCBs in resin pellets from distant islands ranged from 0.1 to 9.9 ng/g-pellet, 1–3 orders of magnitude less than concentrations in resin pellets from industrialized coastal areas [30]. Nevertheless, the sum of PCBs concentration varied considerably among the reviewed studies because of the different number of PCB congeners species analyzed, from 13 [27,29,30,31] to 20 [37] PCB homologs. Such issue can be highlighted also in the study by Dasgupta et al. (2021) [121] who compared the PCBs concentration in MPs collected from different compartments of the marine environment: deep-sea, ocean surface, and beach. The authors found a lower PCBs concentration in deep-sea plastics, but here they analyzed fewer target compounds compared to the other matrices. Both field and laboratory results indicated that there are differences in the sorption capacity of different polymers; for example, PE particles have a higher affinity for PCBs than PP, and, in addition, aged resin pellets contain higher concentrations of PCBs [27,112,122].

#### 3.4.2. Polycyclic Aromatic Hydrocarbons (PAHs)

PAHs are hydrophobic chemicals whose presence in the environment is the result of petroleum leak (i.e., fossil fuels [54]) and pyrolysis (incineration processes [123]). Their interactions with MPs depend on the aging of MPs, color, and polymer type [39,124], but the hydrophobicity of MPs is the main interaction mechanism between MPs and PAHs [125]. 

Almost half of the articles (*n* = 19, Table 3) searched MP-associated PAHs; some of them focused only on PAHs [39,42,43,50], but the majority (15/19, 79%) detected and analyzed them together with other contaminants, such as OCPs, organophosphorus pesticides (OPs), and PCBs. PAH concentrations had a median value of 3595 ng/g, with an extremely high amount of 120,000 ng/g obtained from the MPs collected from Chinese seawater [43].

The concentration of MP-associated PAHs varied greatly among different studies, probably depending on sources of contamination. In a study form Greece [29], the main sources of PAHs were seaport traffic, tire wear, and vehicle exhaust, and the median PAHs concentration was 170 ng/g. However, the authors found that the concentration of PAHs in the resin pellets was lower than expected based on the levels in the water column, probably due to the photodegradation of PAHs. In the southwest coast of Taiwan, Chen et al. (2020) [50] found that PAHs derived mainly from sources associated with petrogenic and vehicles, probably as a result of the exposure of floating MPs to oil spills or tar particles from ships; here, the median PAHs concentration was 456.5 ng/g. In China, Shi et al. (2020) [53] found that foam-associated PAHs levels ranged from 11.2 to 7710 ng/g, and these values were the highest among the hydrophobic chemicals detected on plastic pellets and debris along the studied beaches impacted by petroleum sources. Such a result supports the use of foam for monitoring some MP-associated chemicals, given their higher ability to absorb PAHs compared to widely used resin pellets. Moreover, a lab-scale experiment on the desorption of contaminants from plastic debris to seawater within 24 h [23] showed the ability of MPs to retain PAHs, thus they can also be transported far from their pollution sources. The same authors also found that the average concentration of contaminants in plastic debris was higher than that in sand, thus demonstrating the higher accumulation (or adsorption) degree of contaminants in plastic debris (organic materials) than in sand (inorganic materials).

However, PAHs adsorbed to MPs could derive not only from the surrounding environment but they could also be added during the manufacturing process, especially in the case of PS plastic. In fact, Van et al. (2012) [32] mentioned that unexposed PS foam contained a higher concentration of PAHs than virgin PS pellets, thus the possible sources of PAHs in PS plastics should be carefully considered. 

#### 3.4.3. Organochlorine Pesticides (OCPs)

OCPs are chlorinated compounds used as pesticides in agricultural activities [37,65] and include DDT, HCH, lindane, or dieldrin [126]. DDT decomposes in the environment into dichlorodiphenyl dichloroethylene (DDE) and dichlorodiphenyl dichloroethane (DDD) [36,127]. Today, DDT is banned in several countries due to its persistence and bioaccumulation in the environment [128], but it is currently used at the industrial level in China and as an insecticide against malaria-carrying mosquitoes in South Africa and tropical Asian countries, such as Vietnam [129]. MP-associated OCPs were studied in 17 of the reviewed articles (Table 3 and Appendix A). The most detected contaminants were DDTs (median = 126.9 ng/g, with a maximum concentration of 626 ng/g) and HCHs (median = 5 ng/g, with a maximum concentration of 63.5 ng/g) (Appendix A).

Additionally, in this case, the type of OCPs adsorbed to the MPs is linked to the main pollution sources in the study area. In Hong Kong, most of the DDX in MPs is in the form of DDT, thus supporting a fresh input of DDT into the ocean from anti-fouling paints on fishing vessels and from aquaculture [48]. Additionally, in Vietnam, MPs-associated DDT was present in higher concentrations than its metabolites (DDE and DDD), suggesting that the main source may be the insecticide currently used for malaria control or illegal use [27,38]. Ogata et al. (2009) [27] detected high concentrations of HCH in resin pellets in southern Africa, indicating the continued local use of this pesticide. 

#### 3.4.4. Other Contaminants

In the reviewed papers, some other contaminants have been detected with a lower frequency compared to PCBs, PAHs, and OCPs, and they included mainly plastic additives (e.g., phthalates, brominated flame retardants—BFRs, bisphenol A), but also PFASs and organophosphorus pesticides—OPs (Table 3 and Appendix A). 

BFRs are commonly used in plastic products to prevent fires [40]; they are currently represented by PBDEs, but HBCD have also been used in the past as a flame retardant. MP-associated BFRs were reported in 7 of the reviewed articles [40,45,51,52,57,61,64], with a median concentration of 215.29 ng/g and the highest value of PBDE detected in riverine outlets in China (14,800 ng/g) [51].

Phthalates are used as plasticizers in a variety of plastic products. MP-associated phthalates were reported in five of the reviewed papers [33,44,49,57,58], with the highest concentration (4241.8 μg/g) detected in the plastic samples in Yangtze Estuary wetlands, China [57].

UV filters, UV stabilizers, and antioxidant are frequently employed in the manufacturing of plastics to enhance their physical and chemical properties [41]. Three of the reviewed papers [41,45,62] provided concentration data for the pollutants, and the UV filters were the most abundant, ranging from 1 to 4031 ng/g. However, such chemicals could derive also from sunscreens and other sun care products commonly used by beachgoers [130], and current studies on MPs are not able to determine the origin of the chemicals detected in MPs. 

Other plastic additives include BPA, nonylphenols, and octylphenol, that are extensively used as plastic antioxidants, packaging stabilizers, and reaction reagents in the production of plastics and related products [131]. Their adverse effects on human health are quite well known, since they can behave as endocrine disruptors [132]. Two of the reviewed articles [46,60] investigated their concentration in MPs, and BPA showed the highest levels, with average values of 82.4 and 989 ng/g, respectively.

PFAS are synthetic compounds that are widely used in various industrial and consumer products, such as polymer additives, adhesives, herbicide, and pesticide formulations [133,134,135]. Recent studies have shown that MPs may increase the toxicity of PFASs [136,137]. Such chemicals have been investigated in two of the reviewed studies [35,56], revealing their presence in MPs collected from both beaches (10–180 ng/kg) [35] and estuarine waters (10.3–9.07 × 10^3^ ng/g) [56]. 

OPs (e.g., chlorpyrifos, malathion, and pirimiphos-methyl) are insecticides that have been widely used in agriculture during the last decade [45], and they have been detected in various marine environments [138]. Two of the reviewed articles examined the presence of OPs in association with MPs [45,55] and they found chlorpyrifos as the most representative OPs, with concentrations ranging from 0.5 to 147.61 ng/g in the collected MPs. 

### 3.5. Qualitative Evaluation of the Included Papers 

The score attributed to each reviewed study is reported in Appendix A, that is intended not as a ranking table of study contribution or value, but as a support for designing a protocol for MP-sorbed chemical studies. Overall, the cumulative reliability scores ranged from 10 to 21 with an average of 17.20. The majority of them showed a good reliability (≥75% of the total score), with 32 papers (80%) scoring 16 or higher. The percentages of the score for each criterion were also calculated; they are illustrated in Figure 6. 

The criteria with the highest average score were sampling methods and sample extraction (average = 2.95; 98% of the total score), because most of the studies focused on MPs size > 330 µm, whose separation from the rest of the sample is relatively easy. Not surprisingly, also the detection of chemicals obtained a high score (average = 2.90; 97% of the total score), since such criteria represented the main goal of the reviewed papers. The sample amount (average = 2.15; 72% of the total score) and polymer identification (average = 2.10; 70% of the total score) criteria did not achieve a good reliability, indicating the need for the comprehensive reporting of research information and of instrumental analysis for MPs polymers, respectively. The contamination control (average = 1.35; 45% of the total score) showed the lower scores. Such a result highlights an urgent need for more rigorous quality assurance during the analytical procedures for MP-sorbed chemicals. 

### 3.6. Suggested Protocol for MP Absorbed Chemical Studies

On the basis of the reviewed papers and of the strengths and weaknesses of their methodologies, we proposed a protocol for the collection and analysis of MP-associated chemicals (Table 4). 

## 4. Conclusions

Our systematic review analyzed 40 monitoring studies on MP-related chemicals in the marine environment, considering both the methodology (from sampling to detection of chemicals) and the obtained results in terms of the features of the collected MPs and their associated chemicals. We showed that MPs collected from sand or water can transport various organic pollutants, both acquired from the surrounding environment (PCBs, PAHs, pesticides) and additives that remain incorporated in MPs polymers, mainly flame retardants and phthalates. However, we found considerable heterogeneity in the study design and analytical protocols of different studies, that hampered a precise evaluation on the role of MPs as vehicle of harmful chemicals for human and animal health. For this reason, we developed guidelines for studies on MP-associated chemicals and we can provide the following final recommendations on the basis of the critical issues identified in the reviewed papers:Increase the frequency of sampling campaigns because a single sampling event can lead to inaccurate estimations of MP-associated chemicals level, owing to possible seasonal variation in pollutant inputs;Describe in detail the sampling strategy and sample analysis to allow research replicability;Collect different types of MPs because the ability of MPs to adsorb pollutants varies depending on the size and shape; therefore, sampling only specific shapes and sizes of MPs (such as resin pellets) can lead to biased results on the levels of chemical contamination;Apply contamination control practices that can include both measures to prevent external contamination by MPs and chemicals, and negative controls to analyze simultaneously with the collected environmental samples.

## Figures and Tables

**Figure 1 ijerph-20-04892-f001:**
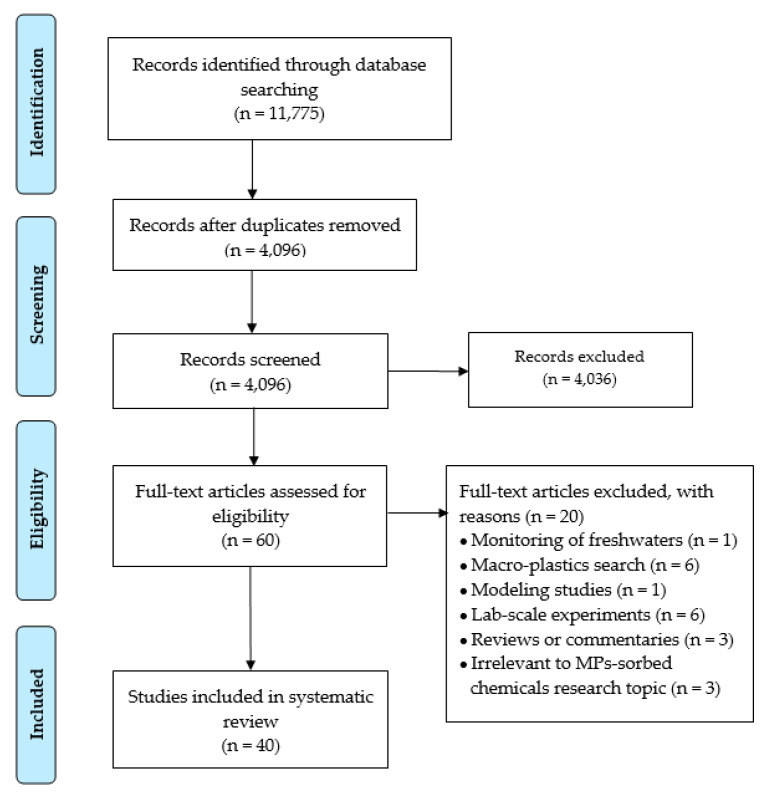
PRISMA flowchart of the study selection process.

**Figure 2 ijerph-20-04892-f002:**
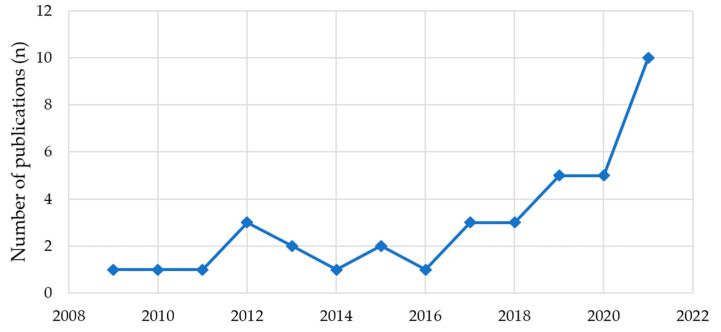
Time trend of publications.

**Figure 3 ijerph-20-04892-f003:**
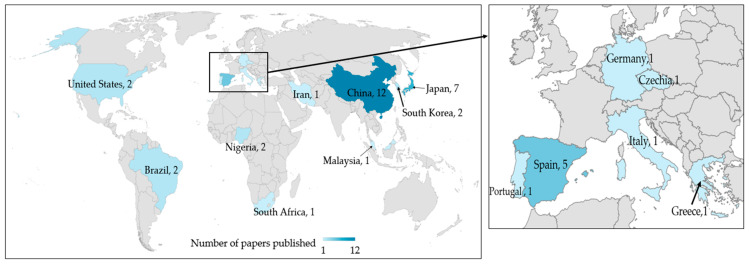
Geographical distribution of the reviewed papers (the number near the country name refers to the number of papers).

**Figure 4 ijerph-20-04892-f004:**
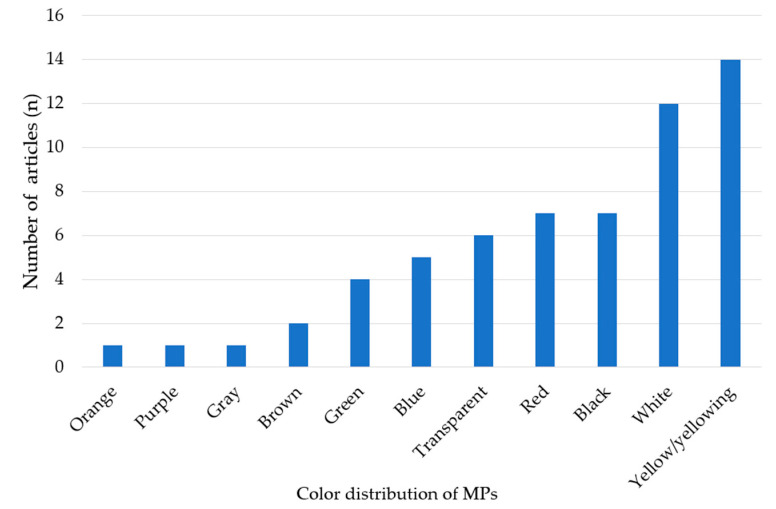
Distribution of MPs color in the reviewed studies (some papers reported more than one color).

**Figure 5 ijerph-20-04892-f005:**
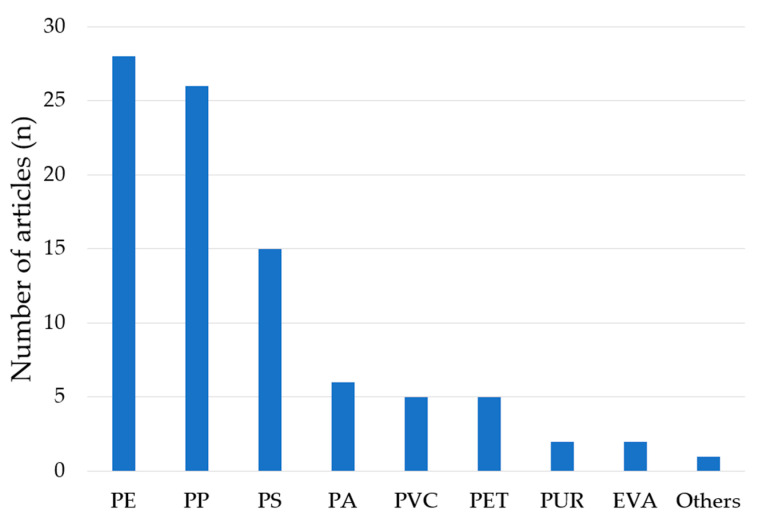
Polymer type reported in the reviewed papers (EVA = ethyl vinyl acetate; PA = polyamide; PE = polyethylene; PET = polyethylene terephthalate; PP = polypropylene; PS = polystyrene; PUR = polyurethane; PVC = polyvinylchloride. Others refer to a very low percentage of polymer types reported in only one article or mentioned in studies only as “others”).

**Figure 6 ijerph-20-04892-f006:**
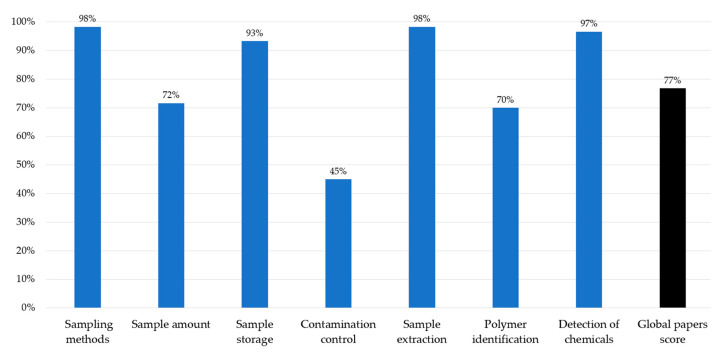
Percentage of score for each criterion. The percentage of the global papers score is calculated summing the single scores obtained from all the reviewed studies.

**Table 1 ijerph-20-04892-t001:** Evaluation criteria description and scoring.

Evaluation Criterion	Description	Score Coding
(i) Sampling methods and sample physical characteristics	Sampling methods include sampling strategy, locations, and depth of the sampling sites, and the tools for the collection. Sample physical characteristics refers to the appearance of MPs (e.g., shape, color, size). They allow a comprehensive interpretation of the study results and possible interferences, such as cross-contamination [69].	3 = Detailed sampling methods and other information (e.g., sampling tools, physical characteristics of the samples); 2 = Brief description of the sampling method and sample size; 1 = Only the sample size was reported, without any information on the sampling method; 0 = No information on sampling.
(ii) Sample amount	The sample amount refers to the number (abundance) of MPs collected from an environmental matrix (e.g., seawater, beach sand).	3 = The abundance of the MPs collected was reported (i.e., number of MP particles per quantity of environmental matrix); 2 = The total amount of the collected environmental matrix is reported (i.e., weight of sand or volume of water); 1 = The total amount for a sample location or just a cursory report of the number of samples analyzed is globally reported; 0 = No information on MPs abundance or sample quantity.
(iii) Sample storage	These criteria refer to storage methods of the samples, either short-term storage (i.e., during the transport from the collection sites to the laboratory) and long-term storage (i.e., time spent in the laboratory before the analysis).	3 = Utilization of non-plastic containers throughout the entire storage process (from sampling to storage in the laboratory); 2 = Use of plastic-related containers in one step of the storage process, but the polymer type of the container was not detected in the analyzed samples; 1 = Use of plastic-related containers in more than two steps of the storage process, but the polymer type of the container was not detected in the analyzed samples; 0 = Use of plastic containers along the storage chain (sample contamination cannot be excluded).
(iv) Contamination control	Cross-contamination is a common problem in MPs research and it can occur throughout the study, including sampling, sample storage, extraction, and analysis. Measures for contamination control include avoiding the usage of plastic-made tools and performing negative controls.	3 = Detailed protocol with cross-contamination precautions and negative controls; 2 = Brief description of contamination prevention measures and negative controls; 1 = Only contamination prevention or negative controls is performed; 0 = No information on contamination prevention measure or negative controls.
(v) Sample extraction	Sample extraction is needed for the separation of MPs from the rest of the environmental matrix. Density separation method is applied in case of samples that are not easily distinguishable to the naked eye owing to their color (e.g., transparent) or size (<1 mm). Such method is usually omitted for resin pellets of foam that can be separate from the environment because their size (>1 mm), color, and shape allow for an easy collection.	3 = Detailed data on the isolation method and the tools or reagents used for the MPs extraction, or studies focused on visually recognizable large microplastics (e.g., resin pellets, foam) and not require reagent for sample separation; 2 = Brief sample separation description in studies focused on tiny MPs (e.g., plastic fibers); 1 = Visual evidence (e.g., photographs) of successful sample separation was provided, yet without any description of the employed sample separation methods; 0 = No information on sample extraction.
(vi) Polymer identification	Polymer identification is a key step for understanding if the collected debris are actually MPs particles or not. If the polymer type is not verified by electronic instrumentation, it is not possible to determine with 100% certainty whether the suspect sample is plastic.	3 = Use of instrumentation to determine the type of polymer; 2 = Polymer type was inferred by referencing published studies from the same location, without direct analysis; 1 = Use of non-electronic instrumental method to determine the polymer type (resin pellets only); 0 = No polymer identification was performed (i.e., purely visual classification).
(vii) Detection of chemicals	The detection of organic chemicals is of critical importance based on the aim of the reviewed studies. Chemicals can be described both qualitatively (i.e., type of chemical compound) and quantitatively (i.e., concentration of each chemical).	3 = Detailed descriptions of instrumental detection methods and quantitative data on MP-sorbed chemicals; 2 = Brief descriptions of instrumental detection methods and quantitative data on MP-sorbed chemicals; 1 = Detailed descriptions of instrumental detection methods and pollutant type, but concentration of the pollutant was not reported; 0 = No information on instrumentation methods.

**Table 2 ijerph-20-04892-t002:** Comparison among analytical instruments for polymer type identification used in the reviewed papers (✔ = yes, ✖ = no).

Type of Analytical Instrument	Identification of Structure and Shape of MPs	Identification of Small MPs (<300 µm)	Sensitivity (Efficiency of MPs Detection)	Cost-Effective	Comments
Infrared spectroscopy (FT-IR, NIR, FIR)	✔	✔/✖	✔/✖	✔	FT-IR is very popular in MPs polymer identification, especially the micro-FT-IR (μ-FT-IR) that can also identify MPs ranged between 20 μm and 300 μm [89]. NIR and FIR spectroscopies are less sensitive compared to FT-IR [98], but their application in MPs search has recently increased as a result of optimization protocols (e.g., enrichment step of MPs prior to the analysis) [99,100,101].
Py-GC/MS	✖	✖	✔	✖	Destructive technique: the samples are thermally decomposed before separation by GC and identification by MS [102]. It is not suitable for the analysis of MPs smaller than 500 μm [78,103] or in the case of samples containing high concentrations of impurities (e.g., soil, sediment) [104].
LC-SEC-HRMS	✖	✔	✔	✖	Destructive technique: the samples are chemically decomposed using solvents (e.g., toluene) before the LC. Particles are able to be identified in the nano-plastic range [63,105]. The type of solvents used for degradation need to be carefully chosen because some polymers (e.g., PE, PP, and PS) are soluble in solvents, especially toluene [63].

FT-IR = Fourier transform infrared; NIR = near-infrared, FIR = far-infrared; Py-GC/MS = thermal decomposition based on pyrolysis gas chromatography/mass spectrometry; LC-SEC-HRMS = liquid chromatography coupled to size exclusion chromatography coupled to high-resolution mass spectrometry.

**Table 3 ijerph-20-04892-t003:** List of chemicals quantitatively measured in MPs in the reviewed studies (some studies identified multiple chemicals).

Compound	No. of Publications	Compound	No. of Publications
PCBs	22	UV filters, UV stabilizers, and antioxidants	3
PAHs	19	PFASs	2
OCPs	17	Other plastic additives (BPA and its analogs, octylphenol, and nonylphenols)	2
BFRs	7		
Phthalates	5	OPs	2

BPA = bisphenol A; BFRs = brominated flame retardants; OCPs = organochlorine pesticides; OPs = organophosphorus pesticides; PAHs = polycyclic aromatic hydrocarbons; PCBs = polychlorinated biphenyls; PFASs = perfluoroalkyl and polyfluoroalkyl Substances.

**Table 4 ijerph-20-04892-t004:** Suggested protocol for MP-associated chemical studies.

1. Sampling methods and sample physical characteristics	Sampling records should be detailed and contain: - Sampling time and location. To obtain more reliable data, it is recommended to select several sampling points and to repeat sampling in a defined timeframe (multiple monitoring campaigns); - Sampling tools and their materials; - Aperture size/mesh size of the sieve or trawl; - Sampling depth; - Sampling strategy (e.g., bulk, volume-reduced, selective); - Sample physical characteristics (e.g., shape, color, size).
2. Sample amount	The amount of sample collected (e.g., dry/wet weight of sediment; volume of water) should be recorded, thus allowing a comparison between the amount/extension of environmental matrix and the abundance of the recovered MPs from that matrix. The abundance of collected MPs should be quantified, choosing a precise unis of measurement (e.g., items/kg, items/m^2^, items/m^3^, and items/L).
3. Sample storage	Samples should be stored in non-plastic containers during sample collection and transfered to the laboratory. Heat or violent shaking should be avoided in order to prevent possible additional degradation (e.g., fragmentation). If samples are stored for long periods before examination, they should be kept frozen at −20 °C.
4. Contamination control	To prevent cross-contamination from external MPs and chemicals, researchers should use plastic-free tools and containers and wear clothing that does not produce plastic fibers (e.g., 100% cotton lab coats). All materials, devices, and laboratory surfaces need to be completely cleaned and rinsed. All materials used in the sample analysis should be kept in a dedicated laboratory space free from plastic stuffs and the equipment should be inspected prior to use. Handling of samples should be done in a clean air biosafety cabin. To check if cross-contamination has happened, negative controls should be performed both for MPs and chemicals in parallel with the sample processing, especially in the case of analysis of chemicals adsorbed by small plastic particles (e.g., plastic fibers < 300 µm). If clean air cannot be guaranteed, a clean Petri dish should be placed next to the sample to check the possible MPs contamination (e.g., 3 replicates of the negative control for each batch of samples). It is recommended to clearly describe the two types of negative controls for evaluating the cross-contamination from: (i) external MPs during sample collection and handling, and (ii) external target chemicals during instrumental analysis of samples.
5. Sample pretreatment	Samples of small plastic particles (e.g., plastic fibers < 300 µm) and with high organic content may require a digestion step to dissolve organic matter. However, before application, it is crucial to verify the possible effect of the solvent used in the digestion step (e.g., 35% hydrogen peroxide) on the recovery of the MP-associated pollutants. Studies on MPs larger than 300 µm could not require sample pretreatment, because residual organic matter not prevent identification of polymers through spectroscopic techniques, such as FT-IR.
6. Sample extraction	A density separation method should be performed to extract the MPs from samples contains interfering impurities. The recommended reagent is saturated NaCl solution because it does not interfere with the subsequent chemical analysis, it is safe for workers and for the environment. Studies on large resin pellet could not require a density separation procedure, because they are easy to distinguish and to recover.
7. Polymer identification	Polymers should be identified through instrumental analysis (e.g., FT-IR, Py-GC/MS). If the entire suspected MPs sample cannot be analyzed, at least a sub-sample of representative particles should be confirmed. In particular, it is recommended to analyze all or at least 50% of the suspected plastic particles if their number is less or greater than 100, respectively. Particle counts and particle size should be reported with confidence intervals and detection limits, respectively.
8. Detection of chemicals	The selection of detection techniques depends on the type of target chemicals. Gas chromatography-based techniques coupled with mass spectroscopy could be a suitable choice for the detection of thermally stable organic pollutants.

FT-IR = Fourier-transform infrared spectroscopy.

## Data Availability

Data are contained within the article or Appendix A.

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
