# Peer review of "Organic Pollutants Associated with Plastic Debris in Marine Environment: A Systematic Review of Analytical Methods, Occurrence, and Characteristics"

_ijerph, 2023, doi:10.3390/ijerph20064892_

Round 1

Reviewer 1 Report

Please, see the attached file

Author Response

Please, see the attached file

Reviewer 2 Report

The manuscript provides a systematic review for the study of compounds sorbed on microplastics, taking into account 40 different papers. The evaluation of sampling methods and protocols was performed according to PRISMA guidelines. The authors, finally, propose a protocol for MPs-sorbed chemical studies.

General considerations: the paper is overall clear and well written. 

In my opinion, the title is misleading: "Emerging organic pollutants adsorbed to plastic debris in marine environment - a systematic review of analytical methods, occurrence, and characteristics" may suggest the revision of literature regarding the analysis of emerging contaminants (EC) sorbed onto microplastics (MP), that is the sampling of MP itself, the methods for extracting EC, the analytical methods for their analysis, their occurrence.

On the contrary, through the paper we find out that most studies dealt with the determination of PAHs, PCBs, Organochlorines, etc., with only a few regarding EC. Consider changing the title to have a better focus of the review.

The review takes into account all steps of MP studies, comparing different methods and highlighting the need of a standardized protocol.

The method proposed for collection and analysis of MPs-associated chemicals is sensible (apart for the treatment with H2O2, topic that will be addressed later), almost obvious (all its steps are evident and straighforward in any sound analytical method), even if it can be agreed upon that the reviewed papers lacked a rigorous method.

Specific comments:

3.1.2 Sample pretreatment (digestion): the authors reports the pre-treatment with hydrogen peroxide to degrade the organic matter prior to other investigation; adding that "the effect of H2O2 pretreatment on the concentrations of MPs-related chemicals has been addressed only by few studies". In spite of that, they suggest in the proposed protocol to use this step in the analytical method.

Please explain the possible effect of this treatment on different chemicals sorbed onto the MP. Would it be possible that some pollutants may degrade?

line 30, correct "tones" with "tons"

Table 1, iv) Contamination control, correct "contamiantion" with "contamination"

line 313: substitute "divided" with "separated"

Overall, the review does not provide a great further improvement in the field of the selected topic.

Reviewer 3 Report

The manuscript represents a comprehensive review of analytical protocols for microplastics-sorbed organic pollutants in marine environment. The topic is interesting, taking into consideration that today, microplastic pollution in the water environment has become an environmental issue of global concern. The cited studies mainly focused on the detection methods of microplastics and organic pollutants in water, as well as their occurrence and effects on water ecosystems. However, our current understanding of microplastics in water is still fragmented, making the manuscript important in the field. In addition, a unified research protocol was suggested for microplastic monitoring studies, fulfilling the need to standardize the procedure.

In my opinion, OCPs, PCBs and PAHs are not considered emerging organic contaminants (EOCs) any more, being detected and analyzed in the last decades. EOCs are organic pollutants that are newly detected or discovered in the environment due to recent analytical developments. 

As key suggestions for future research: the analysis of the environmental risks arising from the accumulation of microplastics in water interacting with organic pollutants; the impact and ecological risks of microplastics on animals, microorganisms, and plants in the water.

Reviewer 4 Report

The review paper entitled "Emerging organic pollutants adsorbed to plastic debris in marine environment - a systematic review of analytical methods, occurrence, and characteristics " written by Hongrui Zhao, Ileana Federigi, Marco Verani and Annalaura Carducci, concerns the comparison and revision of 40 studies on MPs and MPs-sorbed organic contaminants in marine environment using PRISMA guidelines. The same study papers have been then scored for reliability and a protocol for standardizing MPs monitoring studies has been proposed.  

In general, the paper design is quite good, as well as the aim and the meaning of the article, which seem comprehensible resulting really interesting and enlightening. Anyway, the article needs minor revisions.

Materials and methods:

Figure 1: Please improve the quality of the image and the design of the flowchart that appears too basic.

Table 1: Please improve the format of the table trying to reorganize the information.  In some cases, the cells include too many characters and it is difficult to follow.

Line 85-87: It is not clear if papers must meet all criteria or at least one.

Results and Discussion:

Line 114: I would suggest to add a sub heading representing the publication trends section.

Line 135-135: I would have added more information about the sampling area, since it has been demonstrated that different zones in the same location could affect the properties of the sample (e.g. the water content of sediments) and so, its nature, the consequent processing steps and results. Are we talking about beach sediments or the seabed? What about other factors related to sediment samples like sediments density? It could affect the abundance of the collected MPs.

I also suggest to discuss and highlight (e.g. one sentence) the importance of MPs detection and monitoring in river estuarine, since it is mentioned as sampling area too.

Line 163-177: What about data about beach sand? They are not mentioned. Did you consider beach sand as sediment?

Line 168-169: I suggest to be more precise in reporting data from works (e.g. Where did they collected the samples? Different depths of what?)

Line 214-215: It seems that this sentence is not finished. You were reporting an example related to the MPs sorbed chemicals.

Line 309-311: Referring to the statement, why they used it anyway? It requires better explanation.

 Line 321-325: What about good and bad points?

Line 326 (paragraph 3.1.5): It is just a long list of methods. What is lacking is an in-depth discussion on them. Please add some sentences.

Paragraph 3.3.4: Why you decided to report data only about PFAS?

Table 3: You did not report data on sample amount and storage in the text (and so, you also didn´t discuss them) even though they were considered as criteria of reliability. I suggest to add them in the text.

Conclusions:

This section appears too brief and incomplete related to the text. Please add more, better summarizing what has been reported in the whole paper.

Round 2

Reviewer 2 Report

The review has been extensively revised. It could be published, even if some doubts regarding the significant contribution in the field still remain.